# Nucleolar Stress Response via Ribosomal Protein L11 Regulates Topoisomerase Inhibitor Sensitivity of P53-Intact Cancers

**DOI:** 10.3390/ijms232415986

**Published:** 2022-12-15

**Authors:** Yuka Ishihara, Kiyoshiro Nakamura, Shunsuke Nakagawa, Yasuhiro Okamoto, Masatatsu Yamamoto, Tatsuhiko Furukawa, Kohichi Kawahara

**Affiliations:** 1Department of Molecular Oncology, Graduate School of Medical and Dental Sciences, Kagoshima University, Kagoshima 890-8520, Japan; 2Department of Pediatrics, Graduate School of Medical and Dental Sciences, Kagoshima University, Kagoshima 890-8520, Japan

**Keywords:** RPL11, p53, nucleolar stress response, topoisomerase inhibitors, drug sensitivity

## Abstract

Nucleolar stress response is caused by perturbations in ribosome biogenesis, induced by the inhibition of ribosomal RNA processing and synthesis, as well as ribosome assembly. This response induces p53 stabilization and activation via ribosomal protein L11 (RPL11), suppressing tumor progression. However, anticancer agents that kill cells via this mechanism, and their relationship with the therapeutic efficiency of these agents, remain largely unknown. Here, we sought to investigate whether topoisomerase inhibitors can induce nucleolar stress response as they reportedly block ribosomal RNA transcription. Using rhabdomyosarcoma and rhabdoid tumor cell lines that are sensitive to the nucleolar stress response, we evaluated whether nucleolar stress response is associated with sensitivity to topoisomerase inhibitors ellipticine, doxorubicin, etoposide, topotecan, and anthracyclines. Cell proliferation assay indicated that small interfering RNA-mediated RPL11 depletion resulted in decreased sensitivity to topoisomerase inhibitors. Furthermore, the expression of p53 and its downstream target proteins via western blotting showed the suppression of p53 pathway activation upon RPL11 knockdown. These results suggest that the sensitivity of cancer cells to topoisomerase inhibitors is regulated by RPL11-mediated nucleolar stress responses. Thus, RPL11 expression may contribute to the prediction of the therapeutic efficacy of topoisomerase inhibitors and increase their therapeutic effect of topoisomerase inhibitors.

## 1. Introduction

Topoisomerase-targeting therapy is widely used to treat a variety of cancers. DNA topoisomerase is a ubiquitously expressed enzyme that cleaves DNA to untwist and reconnect it [1]. DNA topoisomerase consists of two groups, as follows: class I breaks the single strand DNA of double-stranded DNA, relaxing the DNA one turn at the same time; class II breaks both strands of a double-stranded DNA to unwind it [2]. A covalent bond forms at the cleavage site of the severed DNA strand to form a covalent complex [3]. After the kinks and distortions are corrected, covalent bonds are released and the severed DNA is rejoined [4]. DNA topoisomerase inhibitors bind to topoisomerases that form complexes with DNA, stabilizing their conformation and inhibiting the relaxation of the DNA helical structure [5]. As a result, apoptotic cell death in cancer is induced by the dysregulation of normal DNA synthesis.

Several inhibitors of topoisomerase I and II have been developed. Topotecan is a topoisomerase I inhibitor (Topo I inhibitor) [6]. It has been indicated for the treatment of small-cell lung cancer, ovarian cancer, pediatric malignant solid tumors, and advanced or recurrent cervical cancer [7]. Etoposide is a topoisomerase II inhibitor (Topo II inhibitor), which is a semi-synthetic derivative of 4′-demethylepipodophyllotoxin, a naturally occurring compound [8]. Etoposide appears to be one of the most active drugs used for a variety of cancers, including lung cancer, testicular carcinoma, malignant lymphoma, refractory pediatric neoplasms, hepatocellular, esophageal, gastric, and prostatic carcinoma, ovarian cancer, and chronic and acute leukemia [9]. Another Topo II inhibitor is ellipticine, an alkaloid isolated from the leaves of the evergreen tree *Ochrosia elliptica* [10]. In preclinical experiments and clinical trials, this compound and several of its more soluble derivatives have shown significant antitumor activity. They exhibit tumor-suppressive effects on cancer cell lines of leukemia, brain tumors, breast cancer, and neuroblastoma origin [11]. Epirubicin and doxorubicin are anthracyclines. Anthracyclines are polycyclic aromatic antibiotics extracted from *Streptomyces peucetius* var. *caesius* [12]. Anthracyclines are used against a wide range of cancer types and exert their effects by inhibiting topoisomerase II [13]. They also inhibit DNA synthesis by binding directly to double-stranded DNA [14].

Cellular stress activates the tumor suppressor p53 intracellular signaling pathway. Such stress includes DNA damage, which triggers the ATM-Chk2 and ATR-Chk1 cascades, infection, or oncogene activation, and induces the p19ARF pathway and nucleolar/ribosomal stress [15]. The nucleolar stress response is induced during dysregulation of ribosome biogenesis, including blockage of ribosomal RNA (rRNA) synthesis and generation, caused by actinomycin D, 5-Fluorouracil (5FU), mycophenolic acid, serum starvation, or contact inhibition. This stress initiates a cascade mediated by ribosomal proteins (RPs), particularly ribosomal protein L5 (RPL5), ribosomal protein L11 (RPL11), ribosomal protein L23 (RPL23), and ribosomal protein S7 (RPS7) [16,17,18,19,20]. These RPs are usually located in the nucleolus but are rapidly released into the nucleoplasm upon nucleolar stress, inhibiting MDM2, an E3 ubiquitin ligase that ubiquitinates p53 for proteasomal degradation [21]. Consequently, p53 is stabilized and activated, inducing cell cycle arrest and/or apoptotic cell death. Recently, we demonstrated that the sensitivity of gastric cancer cells to an anticancer drug 5-FU is affected by the RPL11-mediated nucleolar stress response [22]. In addition, actinomycin D sensitivity in colon cancer and osteosarcoma cells is regulated by the nucleolar stress response [23]. These observations suggest that nucleolar stress response may be involved in the regulation of cancer sensitivity to certain drugs. However, the relationship between the nucleolar stress response and topoisomerase inhibitor sensitivity in human cancers is unknown.

In the present study, using rhabdomyosarcoma and rhabdoid tumor cell lines that are sensitive to the nucleolar stress response, we investigated the effect of RPL11-mediated nucleolar stress response on sensitivity to topoisomerase inhibitors. Our results reveal that the nucleolar stress response via RPL11 regulates the sensitivity of these cancer cells to topoisomerase inhibitors.

## 2. Results

### 2.1. Sensitivity to Topoisomerase Inhibitors Was Regulated via RPL11-Mediated Nucleolar Stress Response

To examine whether nucleolar stress response via RPL11 may be associated with topoisomerase inhibitor sensitivity, we conducted 3-(4,5-dimethylthiazol-2-yl)-2,5-diphenyltetrazolium bromide (MTT) cell growth assay using p53 wild-type malignant rhabdoid tumors with or without RPL11 knockdown. The siRNA-mediated RPL11 knockdown in p53 wild malignant rhabdoid tumor JMU-RTK-2 cells decreased sensitivity to all topoisomerase inhibitors, including ellipticine (Topo II inhibitor), etoposide (Topo II inhibitor), and topotecan (Topo I inhibitor), compared with scramble siRNA-transfected cells (Figure 1A–C). Siremadlin is an MDM2 inhibitor that directly activates the p53 pathway via MDM2 inhibition without affecting the nucleolar stress response. Since the siremadlin is a negative control drug that activates p53 pathway and reduces the cell survivability regardless of RPL11-mediated nucleolar stress response, it is expected that reduced nucleolar stress response via RPL11 knockdown would not alter p53 pathway activation and drug sensitivity by this drug. Indeed, the sensitivity of JMU-RTK-2 cells to siremadlin was not significantly affected by RPL11 knockdown (Figure 1D). A similar resistance to the three topoisomerase inhibitors was observed in p53 wild-type rhabdomyosarcoma RMS-YM cells transfected with RPL11 siRNA (Figure 2A–C), whereas there was no significant difference in sensitivity to the MDM2 inhibitor siremadlin between siRPL11- or control siRNA-transfected RMS-YM cells (Figure 2D). Nucleolar stress response induced by ellipticine, etoposide, and topotecan, but not by DMSO and siremadlin, was confirmed nucleolar disassembly judged by the diffused distribution of the nucleolar marker protein nucleophosmin in the nucleus (Appendix A). These results suggested that nucleolar stress response was involved in the sensitivity of p53-intact cancer cells to topoisomerase inhibitors.

### 2.2. Topoisomerase Inhibitors Induced P53 Pathway Activation via Nucleolar Stress Response

To investigate whether nucleolar stress response is associated with topoisomerase inhibitor-mediated control of the p53 pathway, we conducted biochemical analyses of p53 and its downstream targets MDM2 and P21. Ellipticine-induced upregulation of p53, MDM2, and p21 in JMU-RTK-2 cells was greatly reduced in RPL11-knockdown cells (Figure 3). The modestly decreased expression of p53 and its targets in cells transfected with RPL11 siRNA #2 was observed in cells treated with etoposide and topotecan (Figure 3). Conversely, MDM2 inhibitor treatment increased the expression of p53 and its targets; however, there was no significant difference with or without RPL11 knockdown (Figure 3). RPL11 expression in RPL11 siRNA-transfected cells was significantly lower than that in negative control siRNA-transfected cells (Figure 3 and Figure 4), suggesting that RPL11 siRNA-mediated knockdown was successful under our experimental conditions. Furthermore, a similar effect was observed on the p53 pathway upon RPL11 knockdown by topoisomerase inhibitors in RMS-YM cells (Figure 4). These observations suggested that topoisomerase inhibitors might be involved in the regulation of the p53 pathway via nucleolar stress response.

### 2.3. Anthracyclines Sensitivity Was Also Affected by the Nucleolar Stress Response

We further investigated whether topoisomerase inhibitor sensitivity is more generally involved in nucleolar stress response using anthracycline agents. MTT assay revealed that RPL11 knockdown significantly decreased the sensitivity of JMU-RTK-2 cells to epirubicin and doxorubicin compared with the negative control (Figure 5A,B). The activation of the p53 pathway by epirubicin and doxorubicin was faintly reduced upon RPL11 siRNA#2 transfection (Figure 5C), similar to that observed when cells were treated with etoposide and topotecan (Figure 3). Nucleolar stress response induced by epirubicin and doxorubicin was confirmed by nucleolar disassembly (Appendix A). These results suggested that sensitivity to anthracycline, in addition to ellipticine, etoposide, and topotecan, might be regulated by the nucleolar stress response.

## 3. Discussion

In the present study, we investigated whether the nucleolar stress response via RPL11 is involved in the sensitivity of cancer cells to topoisomerase inhibitors. The results showed that cell viability after treatment with five topoisomerase inhibitors was significantly increased upon RPL11 knockdown in two different cancer cell lines carrying the wild-type *TP53*. Further biochemical analysis revealed that activation of the p53 pathway was decreased upon RPL11 depletion in cell lines treated with topoisomerase inhibitors. Taken together, these results provide evidence that the RPL11-mediated nucleolar stress response affects the sensitivity of cancer cells to topoisomerase inhibitors by regulating p53 signaling (Figure 6).

Several reports have suggested that topoisomerase inhibition induces dysregulation ribosomal RNA (rRNA) synthesis. For instance, loss of topoisomerase I generates truncated pre-r-RNA, and loss of both topoisomerase I and II blocks pre-rRNA synthesis in yeast [24]. The selective inhibition of topoisomerase I activity by camptothecin leads to the inhibition of rRNA synthesis in vitro [25]. Furthermore, the inhibition of pre-rRNA synthesis was observed in the neocortical neurons of adult rats after intracarotid injection of etoposide [26]. A previous report demonstrated that topoisomerase IIα is a component of the initiation-competent RNA polymerase I complex and promotes RNA polymerase I-driven rRNA transcription [27], suggesting the relationship between the perturbation of RNA polymerase I-driven rRNA transcription and topoisomerase inhibition. In addition, RNA polymerase I inhibitor CX-5461 exerts its cytotoxic activity through topoisomerase II inhibition [28]. Growing evidence suggests that topoisomerase inhibition might be associated with the dysregulation of rRNA generation, which induces nucleolar stress. Since the dysregulation of rRNA synthesis causes a nucleolar stress response, these observations support our present data that topoisomerase inhibitors are involved in the induction of the nucleolar stress response.

Topoisomerase inhibitors are clinically used to treat a variety of cancers, including hematological malignancies and solid tumors. However, the therapeutic effectiveness of topoisomerase inhibitors is often limited by resistance. It is now apparent that the resistance mechanism of topoisomerase inhibitors involves removing the drug or the drug target and/or changing cellular response to the drug or interfering with DNA damage detection. Mechanistically, topoisomerase inhibitor resistance is associated with the decreased expression of topoisomerases or mutations that reduce the affinity to inhibitors [2,29]. Other resistance mechanisms include the increased expression of multi-drug resistant proteins such as membrane transporters, which are responsible for drug efflux, reducing intracellular drug concentrations [30]. Another important layer of topoisomerase inhibitor resistance is the p53 signaling pathway. Topoisomerase inhibitors induce DNA damage, p53 upregulation, and apoptosis. Indeed, *TP53* mutation in patients with breast cancer is associated with primary resistance to doxorubicin therapy [31]; the loss of functional p53 protein confers resistance to etoposide in neuroblastoma and glioma cells [32,33]. Moreover, transfection with wild-type p53 sensitizes soft tissue sarcoma cells to doxorubicin [34]. Considering these previous observations and the present study, it is suggested that p53 signaling regulates topoisomerase inhibitor sensitivity, supporting our results that p53 regulation via the nucleolar stress response is involved in topoisomerase inhibitor sensitivity.

RPL11 expression could be useful as a biomarker for assessing the efficacy of topoisomerase inhibitor treatment. For patients with reduced RPL11 expression, the administration of agents whose actions are not dependent on the nucleolar stress response may be an effective strategy to increase therapeutic efficacy. By contrast, for patients without reduced RPL11 expression, topoisomerase inhibitors may be effective in cancer patients. Dopeso et al. reported that the DNA repair factor Aprataxin (APTX) regulates the response to the Topo I inhibitor irinotecan in metastatic colorectal cancer, with lower protein expression associated with a longer survival period [35]. Further research is needed; however, cancer patients with both high RPL11 and low APTX expression may display higher treatment sensitivity to topoisomerase inhibitors than those with either high RPL11 or low APTX expression alone. This may contribute to a more accurate prediction of topoisomerase inhibitor sensitivity in cancers. In addition, RPL11 expression may also be a potential therapeutic target. Resistance to topoisomerase inhibitors is a problem that must be solved urgently. As a decreased expression of RPL11 leads to reduced sensitivity to topoisomerase inhibitors, the development of drugs that increase RPL11 expression could contribute to increasing the efficacy of topoisomerase inhibitors and overcoming resistance in cancers.

Although drug sensitivity assays have shown that the sensitivity to all topoisomerase inhibitors examined in the present study is regulated by the RPL11-mediated nucleolar stress response, the strength of the effect on p53 signaling varied greatly from drug to drug. For instance, in the case of ellipticine treatment, a considerable reduction in the p53 pathway activity by RPL11 knockdown is commonly observed in two different cell lines, whereas the reduced effect on p53 pathway activation by RPL11 knockdown is weaker in the treatment with etoposide, topotecan, doxorubicin, and epirubicin. Topoisomerase inhibitors cause DNA damage, inducing p53 accumulation [36]. It has been reported that RPL11 binding-deficient Mdm2C305F mutant knock-in mice retain a normal p53 response to DNA damage but fail to stabilize p53 in response to nucleolar stress, suggesting that RPL11-mediated nucleolar stress response may be independent of DNA damage stress response [37]. The accumulation of p53 induced by DNA damage due to topoisomerase inhibitor treatment is distinct from and, at least in part, independent of RPL11-mediated nucleolar stress response. Therefore, different effects on the p53 pathway between ellipticine and other topoisomerase inhibitors may reflect the proportion of DNA damage and nucleolar stress response induction by the type of topoisomerase inhibitors. The p53 independent nucleolar stress response mechanism may also be associated with the differential effect of topoisomerase inhibitors on the p53 pathway [38]. RPL11 regulates p53-independent nucleolar stress response via Myc oncoprotein [37]. For instance, RPL11 blocks c-Myc function and expression by directly binding both c-Myc and its mRNA, inducing cell growth suppression independent of p53 [39,40]. Topoisomerase inhibitor-induced nucleolar stress response mechanisms may include p53-dependent and p53-independent mechanisms, which may more strongly affect sensitivity depending on the type of drug. Sloan et al. reported that RPL5 and RPL11 regulate p53 from the context of a ribosomal subcomplex, the 5S ribonucleoprotein particle (RNP), upon nucleolar stress [41]. RPL5, along with RPL11, is also expected to be involved in the regulation of topoisomerase inhibitor sensitivity. Furthermore, no in vivo or tumor patient studies have shown a relationship between topoisomerase inhibitors and nucleolar stress responses. A complete understanding of this requires further investigation of these subjects.

## 4. Materials and Methods

### 4.1. Cell Culture and Reagents

The human malignant rhabdoid tumor cell line JMU-RTK-2 was obtained from the JCRB cell bank and cultured in Dulbecco’s modified Eagle’s medium supplemented with 5% fetal bovine serum. The rhabdomyosarcoma cell line RMS-YM was obtained from RIKEN BRC cell bank and cultured in RPMI-1640 supplemented with 10% fetal bovine serum, 20 mM-HEPES, and 0.1 mM None-essential amino acids (FUJIFILM Wako Pure Chemical Corporation). All the cells were maintained at 37 °C in a humidified atmosphere containing 5% CO_2_. Topoisomerase inhibitors, etoposide, doxorubicin, and epirubicin were purchased from FUJIFILM Wako Pure Chemical Corporation (Osaka, Japan), ellipticine was purchased from Merck (DS, Darmstadt, Germany), topotecan hydrochloride hydrate was purchased from Sigma Aldrich (St Louis, MO, USA), and siremadlin was purchased from Medchemexpress (Monmouth Junction, NJ, USA).

### 4.2. SiRNA Transfection

We transfected cells with siRNA oligonucleotides in Screen Fect Dilution Buffer (FUJIFILM Wako Pure Chemical Corporation) using Screen Fect siRNA (FUJIFILM Wako Pure Chemical Corporation), according to the manufacturer’s protocol. SiRNA sequences were as follows: RPL11 siRNA#1, 5′-GGUGCGGGAGUAUGAGUUA-3; RPL11 siRNA#2, 5′-CAAAUAAAUUCCCGUUUCUAUCC-3; scramble siRNA, 5′-UUCUCCGAACGUGUCACGU-3′.

### 4.3. MTT Assay

The sensitivity of the cells to drugs was estimated using the MTT colorimetric assay. JMU-RTK-2 cells were seeded at 15,000 cells per well, and RMS-YM cells at 10,000 cells per well into a 96-well plate and the MTT assay was performed as described previously [22].

### 4.4. Immunoblotting

After treating the cells with a concentration of the drug similar to the GIC_90_ value, cells were lysed in a buffer containing 20 mM Tris-HCl (pH 7.5), 150 mM NaCl, 1 mM sodium vanadate, 1 mM EDTA, 50 mM NaF, 1% Triton X-100, and protease inhibitor cocktail (Nacalai Tesque, Kyoto, Japan) with sonication at 4 °C. Protein lysate was resolved via sodium dodecyl sulfate-polyacrylamide gel electrophoresis (SDS-PAGE) and proteins were transferred to a polyvinylidene difluoride membrane filter (Immobilon P; Millipore, Burlington, MA, USA). The blots were subjected to immunoblotting analysis using primary antibodies against the following: MDM2 (1:500 dilution; sc-965, SANTA CRUZ BIOTECHNOLOGY, INC, Santa Cruz, CA, USA), RPL11 (1:1000 dilution; 3A4A7, Invitrogen, Waltham, MA, USA), p53 (1:400 dilution; sc-126, SANTA CRUZ BIOTECHNOLOGY, Inc., Dallas, TX, USA), P21 (1:3000 dilution; ab109520, Abcam, Cambridge, UK), and Actin (1:3000 dilution; PM053, MEDICAL & BIOLOGICAL LABORATORIES Co., LTD, Tokyo, Japan). Immunoreactive band signal intensities were detected using LuminoGraph II (ATTO, Tokyo, Japan).

### 4.5. Immunofluorescence Analysis

After treating the cells with the drug at a concentration similar to the GIC_90_ value, cells were fixed in 4% formaldehyde. Cells were then dehydrated with cold methanol in acetone (1:1). After washing extensively with phosphate-buffered saline, fixed cells were stained with anti-nucleophosmin antibody (SIGMA Aldrich, St Luis, MO, USA) and visualized with Alexa Fluor^TM^ 594-conjugated secondary antibody (Thermo Fisher Scientific, Waltham, MA, USA). DNA was counterstained with DAPI (DOJINDO, Kumamoto, Japan). The immunofluorescent signal was visualized using LSM900 confocal microscopy (Zeiss, Oberkochen, Germany).

### 4.6. Statistical Analyses

Data are presented as the mean ± standard deviation. Differences between multiple groups were determined using two-way ANOVA followed by Dunnett’s post hoc test. The analyses were performed using GraphPad Prism software (version 9.4.1; GraphPad Software Inc., San Diego, CA, USA). Differences below the probability level of 0.05 were considered statistically significant.

## 5. Conclusions

In conclusion, we identified RPL11-mediated nucleolar stress response as one of the mechanisms of action of topoisomerase inhibitors. Our work also shows that RPL11-mediated nucleolar stress response is crucial for sensitivity to topoisomerase inhibitors, suggesting that RPL11 expression may be a potential biomarker for predicting topoisomerase inhibitor sensitivity in cancer. Since RPL11 expression regulates sensitivity to topoisomerase inhibitors, drugs that increase RPL11 expression may contribute to increasing the therapeutic effect of topoisomerase inhibitors and overcoming resistance. Nucleolar stress response may be a new target for therapeutic strategy against topoisomerase inhibitors treatment with cancers.

## Figures and Tables

**Figure 1 ijms-23-15986-f001:**
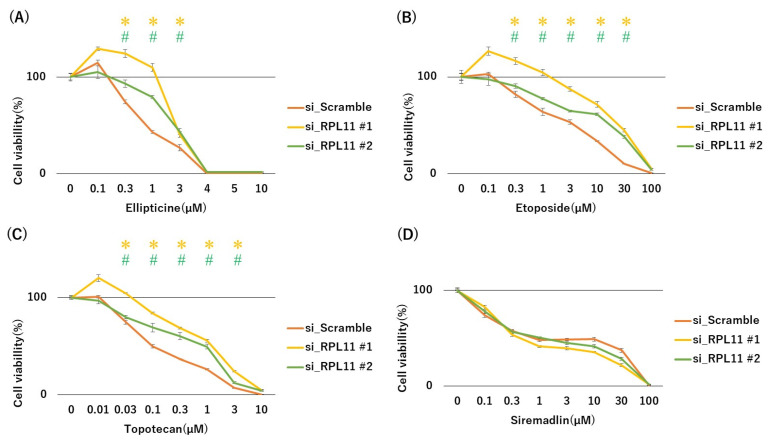
RPL11-mediated nucleolar stress response regulates sensitivity to topoisomerase inhibitors in p53 wild-type malignant rhabdoid tumor JMU-RTK-2 cells. Cells were transfected with scramble, RPL11#1, and RPL11#2 siRNAs and treated with the indicated concentrations of drugs ((**A**) Ellipticine, (**B**) Etoposide, (**C**) Topotecan, (**D**) Siremadlin). After 3 days of culture, the number of surviving cells was measured using the MTT assay. * *p* < 0.05 vs.RPL11#1 siRNA group; ^#^ *p* < 0.05 vs. RPL11#2 siRNA group. Results represent independent five experiments.

**Figure 2 ijms-23-15986-f002:**
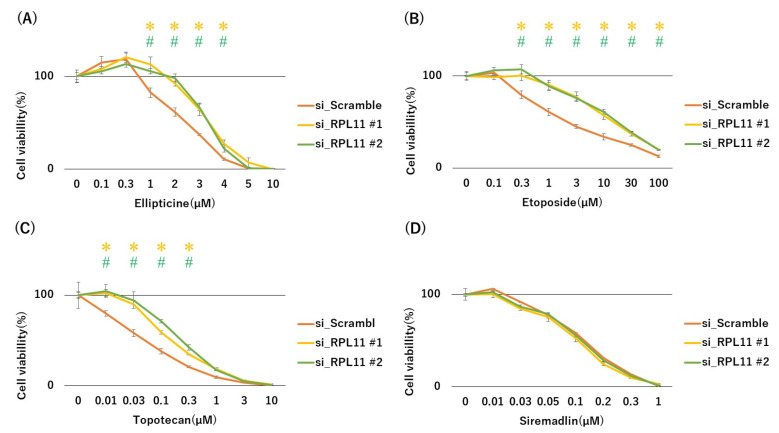
Effect of nucleolar stress response on topoisomerase inhibitor-mediated survival in the p53 wild-type rhabdomyosarcoma cells. RMS-YM cells were transfected with scramble, RPL11#1, and RPL11#2 siRNAs and treated with the indicated concentrations of drugs ((**A**) Ellipticine, (**B**) Etoposide, (**C**) Topotecan, (**D**) Siremadlin). After 3 days of culture, the number of surviving cells was measured using the MTT assay. * *p* < 0.05 vs.RPL11#1 siRNA group; ^#^ *p* < 0.05 vs. RPL11#2 siRNA group. Results represent independent five experiments.

**Figure 3 ijms-23-15986-f003:**
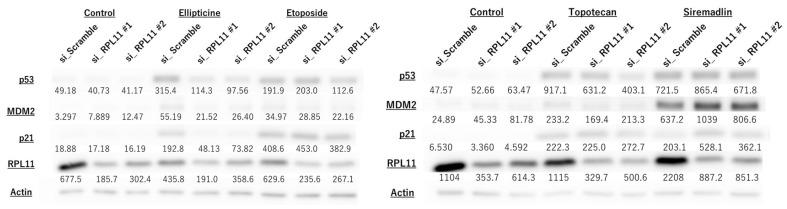
Reduced nucleolar stress response suppresses topoisomerase inhibitor-mediated activation of the p53 pathway in malignant rhabdoid tumor JMU-RTK-2 cells. Cells were transfected with scramble, RPL11#1, and RPL11#2 siRNA and treated with drugs (3 µM Ellipticine, 30 µM Etoposide, 1 µM Topotecan, 1 µM Siremadlin). After 24 h of culture, the cell lysate was subjected to immunoblotting with antibodies against p53, MDM2, p21, RPL11, and actin. Actin was used as a loading control. The numbers below the bands indicate the ratio of the protein/actin expression determined using LuminoGraphII Image Analyzer with CS Analyzer 4 Software. Results represent independent four experiments.

**Figure 4 ijms-23-15986-f004:**
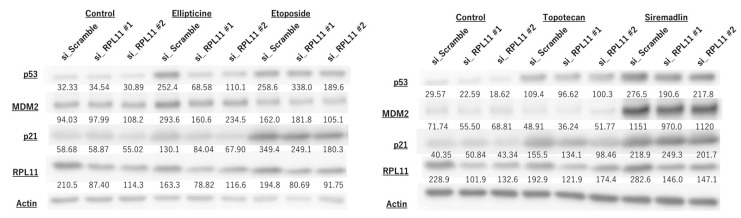
Nucleolar stress response regulates p53 pathway activation by topoisomerase inhibitors in rhabdomyosarcoma cells. RMS-YM cells were transfected with scramble, RPL11#1, and RPL11#2 siRNA and treated with indicated drugs (5 µM Ellipticine, 30 µM Etoposide, 1 µM Topotecan, 0.3 µM Siremadlin). After 4 h of culture, the cell lysate was subjected to immunoblotting with antibodies against p53, MDM2, p21, RPL11, and Actin. Actin was used as a loading control. The numbers below the bands indicate the ratio protein/actin expression determined using LuminoGraphII Image Analyzer with CS Analyzer 4 Software. Results represent independent four experiments.

**Figure 5 ijms-23-15986-f005:**
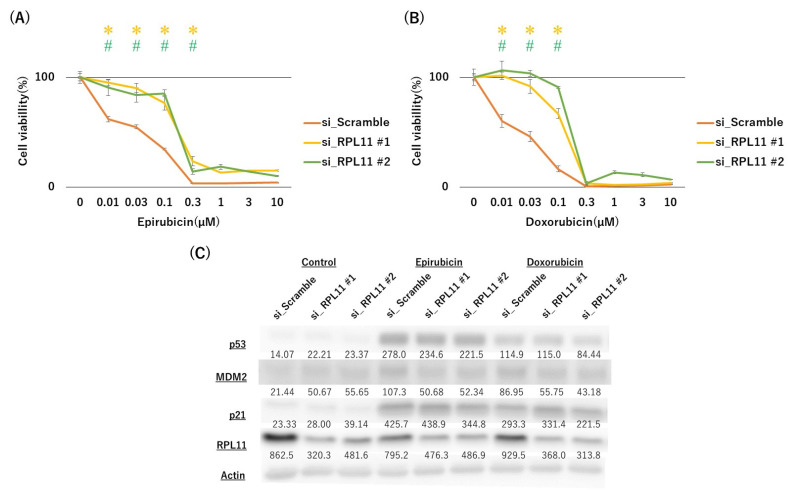
Effect of nucleolar stress response on sensitivity to anthracycline and p53 pathway activation by anthracycline drugs. JMU-RTK-2 cells were transfected with scramble, RPL11#1, and RPL11#2 siRNAs and treated with the indicated concentrations of drugs ((**A**) Epirubicin and (**B**) Doxorubicin). After 3 days of culture, the number of surviving cells was measured using the MTT assay. (**C**) JMU-RTK-2 cells were transfected with scramble, RPL11#1, or RPL11#2 siRNA and treated with 0.3 µM Epirubicin, or 0.1 µM Doxorubicin. After 24 h of culture, the cell lysate was subjected to immunoblotting with antibodies against p53, MDM2, P21, RPL11, and Actin. Actin was used as a loading control. The numbers below the bands indicate the ratio of protein/actin determined using LuminoGraphII Image Analyzer with CS Analyzer 4 Software. * *p* < 0.05 vs.RPL11#1 siRNA group; ^#^ *p* < 0.05 vs. RPL11#2 siRNA group. Results represent independent three experiments.

**Figure 6 ijms-23-15986-f006:**
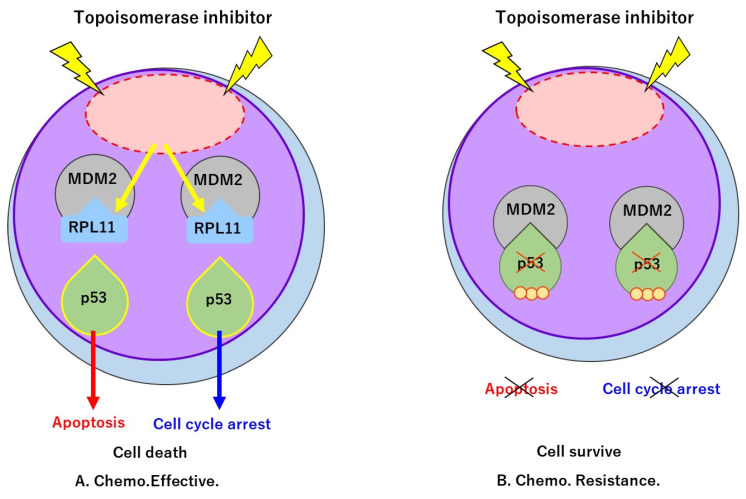
Nucleolus stress response may be one of the determinants of topoisomerase inhibitor sensitivity in cancer. (**A**) When RPL11 is normally expressed, topoisomerase inhibitors induce RPL11 to bind to MDM2 and block ubiquitination-dependent degradation of p53. Consequently, cancer cells inhibit cell growth and induce apoptosis. (**B**) When the expression of RPL11 is low, RPL11 is unable to induce sufficient inhibition of MDM2, resulting in the destabilization of p53 and resistance to topoisomerase inhibitors in cancer cells.

## Data Availability

Not applicable.

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
