# Peer review of "Nucleolar Stress Response via Ribosomal Protein L11 Regulates Topoisomerase Inhibitor Sensitivity of P53-Intact Cancers"

_ijms, 2022, doi:10.3390/ijms232415986_

Round 1

Reviewer 1 Report

While the focus of this manuscript that topoisomerase inhibitor resistance is at least in part linked to the NSR/RPL11 abundance is interesting, however the idea that a wide variety of cellular stress induces/are dependent on a functional NSR is not new. The conclusion of this study that the therapeutic efficacy of topoisomerase inhibitor treatment depends on  RPL11/activation of the NSR needs to be supported by additional experimental replication and new experiments.

The reduction in p53 stabilization in RPL11KD cells treated with Etoposide, Topotecan (RMS-YM cells) and both anthracyclines is not convincingly supported by the data presented in this manuscript. The lack of significant reduction of p53 raises the question whether the therapeutic response is truely due p53 dependent NSR activation. It would be interesting to analyse ATM-ATR DNA damage signalling in this context, is ATM-ATR DNA damage signalling dependent on RPL11?

The RPL11KD efficiency is variable and in some cases only ~50%. It appears that all western blot experiments are n=1.  For all experiments the number of replicates is not clear.

It is not clear what the numbers in Figure 3,4,5 represent, how have the bands been quantified? The quality of these figures need to be significantly improved.

It is unclear how the drug concentrations have been selected for the WB experiments, this should be consistently the GIC50 or 90 for the evaluation of NSR signalling activation.

Have the authors investigated the role of RPL5?  Both RPL5 and RPL11 are components of the 5S-RNP and described to bind to MDM2 in response nucleolar stress.  

Some readout for the perturbation of ribosome biogenesis should be included i.e. measuring rRNA synthesis or processing or changes in nucleolar morphology.

The authors do not mention/discuss the role of Topo2 at the rDNA promoter, the role of RPL5 and CX-5461 as Top II inhibitor.  

Minor points:

P53 should be p53

Ribosomal biogenesis should be ribosome biogenesis

 There are some typos throughout the manuscript.  

Author Response

1) The reduction in p53 stabilization in RPL11KD cells treated with Etoposide, Topotecan (RMS-YM cells) and both anthracyclines is not convincingly supported by the data presented in this manuscript. The lack of significant reduction of p53 raises the question whether the therapeutic response is truely due p53 dependent NSR activation. It would be interesting to analyse ATM-ATR DNA damage signalling in this context, is ATM-ATR DNA damage signalling dependent on RPL11?

We wish to express our gratitude to Reviewer#1 for their valuable suggestions. It has been reported that RPL11 binding-deficient Mdm2C305F mutant knock-in mice retain a normal p53 response to DNA damage but fail to stabilize p53 in response to nucleolar stress, suggesting that RPL11-mediated nucleolar stress response may be independent of DNA damage stress response (Holmberg Olausson K.H et al., Cells. 2012,1(4)774-98). Therefore, p53 accumulation induced by DNA damage upon topoisomerase inhibitor treatment is distinct from and, at least in part, independent of RPL11-mediated nucleolar stress response. This may be why RPL11 depletion could not completely reverse p53 accumulation caused by certain topoisomerase inhibitors. We have addressed this issue in the discussion section on pages 8-9, lines 257-263.

2) The RPL11KD efficiency is variable and in some cases only ~50%. It appears that all western blot experiments are n=1.  For all experiments the number of replicates is not clear.

Thank you for your comment. We apologize for the missing description of the number of replicates in each experiment. I agree that the knockdown efficiency against RPL11 varies from each blot, as you have pointed out. This may be dependent on the cellar conditions in each experiment. However, identical results in all experiments were obtained after multiple independent trials ensuring reproducibility, and a representative blot is shown. For your confirmation, we have added the data for different western blotting analyses. Please refer to page 3, line 113; page 4, line 120; page 5 145-146; page 5 lines 155-156; page 6, line 180.

3) It is not clear what the numbers in Figure 3,4,5 represent, how have the bands been quantified? The quality of these figures need to be significantly improved.

Thank you for your comment. We apologize for the missing description of the numbers in Figures 3, 4, and 5. The numbers in the figures show the relative protein band intensity normalized to the actin band intensity of an identical sample. We have addressed this issue in the revised Figures 3, 4, and 5C on page 5, lines 144-145; page 5, lines 154-155; page 6, lines 178-179.

4) It is unclear how the drug concentrations have been selected for the WB experiments, this should be consistently the GIC50 or 90 for the evaluation of NSR signalling activation.

Thank you for your comment. We agree with the reviewer that we should have conducted western blotting experiments using GIC50 or 90. Since treatment at IC90 may cause cell death depending on the drugs and cell type, we chose a concentration close to the IC90 value at which cells can survive. We have addressed this issue in the Material and Methods section on page 9, line 305; page 10, line 320.

5) Have the authors investigated the role of RPL5?  Both RPL5 and RPL11 are components of the 5S-RNP and described to bind to MDM2 in response nucleolar stress.  

We thank the reviewer for their insightful suggestion. Solan KE et al. reported that RPL5 and RPL11 regulate p53 from the context of a ribosomal subcomplex, 5S ribonucleoprotein particle (RNP), upon nucleolar stress (Sloan K.E et al., Cell Rep. 2013, 5(1)237-47). As the reviewer pointed out, 5S-RNP may be associated with the regulation of sensitivity to topoisomerase inhibitors via RPL11-mediated nucleolar stress response. This is one of the important themes of our project. We are planning to analyze whether RPL5 is included in this mechanism in future studies. We have addressed this issue in the Discussion section on page 9, lines 273-276.

6) Some readout for the perturbation of ribosome biogenesis should be included i.e. measuring rRNA synthesis or processing or changes in nucleolar morphology.

We agree with the reviewer’s suggestion. As requested by the reviewer, we conducted immunofluorescence analysis to check nucleolar morphology. As a result, the nucleolar marker nucleophosmin was disassembled into the nucleoplasm in both cell-lines treated with all topoisomerase inhibitors, showing nucleolar disassembly. These results suggested that topoisomerase inhibitors may induce nucleolar stress response. We have included this data in revised Supplemental Figure 1 and addressed this issue in the Results section on page 3, lines 102-105 and pages 5, lines 165-167, in the Materials and methods section on page 10, lines 315-327, as well as in the Supplementary Material section on page10, lines345-349.

7) The authors do not mention/discuss the role of Topo2 at the rDNA promoter, the role of RPL5 and CX-5461 as Top II inhibitor.  

We thank the reviewer for their constructive suggestion. A previous report indicated that topoisomerase IIα is a component of the initiation-competent RNA polymerase I complex and promotes RNA polymerase I-driven rRNA transcription (Ray S et al., Nat Commun., 2013, 4, 1598), suggesting the relationship between perturbation of rRNA transcription and topoisomerase inhibition. In addition, RNA polymerase I inhibitor CX-5461 exerts its cytotoxic activity through topoisomerase II inhibition (Bruno PM et al., Proc Natl Acad Sci U S A. 2020 117(8)4053-4060). These observations provide mechanistic insight into how topoisomerase inhibitors cause nucleolar stress response, supporting our results. We have addressed this issue in the Discussion section on pages 7-8, lines 204-209.

8) Minor point : P53 should be p53. Ribosomal biogenesis should be ribosome biogenesis. There are some typos throughout the manuscript.

We are grateful to the reviewer for bringing this to our attention. We have corrected “P53” as “p53” throughout the revised manuscript. We have now changed “Ribosome biogenesis” in the Abstract section on page 1, line 13 and in the Introduction on page 2, lines 65. We have corrected ellipticine in the Results section on page 4, line 125.

Reviewer 2 Report

The paper by Ishihara et al. shows that depletion of RPL11 by RNA interference (nucleolar stress) causes a decreased sensitivity to topoisomerase inhibitors that are often used in cancer therapy.  The read-out to measure decreased sensitivity to the topoisomerase inhibitors is increased cell viability (Figures 1 and 2).

My understanding is the authors then show in Figures 3 and 4 how the loss of RPL11 by RNAi suppresses the expression of p53 caused presumably by DNA damage resulting from the inhibition of topoisomerase.

Critiques.

1) I have to admit I get lost in the descriptions of Figure 3.  The Fig. 3 legend begins by saying “Reduced nucleolar stress response suppresses topoisomerase inhibitor mediated activation of the p53 pathway…”  Should this read “Enhanced nucleolar stress response suppresses inhibitor mediated activation of p53…”? The nucleolar stress is induced by depleting RPL11.  When more RPL11 is depleted (enhancing nucleolar stress), there is a corresponding loss of p53 (a suppression of the p53 activation pathway).  If this is the correct interpretation, then the text and Fig. 3 legend need to be re-written to make it more clear.

2) RPL11 is known to interact with Myc (e.g., Olausson et al., Cells 2012 1, 774-798; doi:10.3390/cells1040774), so could the effects of depleting RPL11 be related to enhanced Myc activities?  The western blots in Fig. 3, 4, and 5 should also show Myc and perhaps protein products expressed from known Myc target genes.

3) With the depletion of RPL11, is there an adverse effect on overall ribosome function and protein synthesis in general?

4) There are several minor discrepancies. For example, lines 89 and 90 and again in 119 and 120 the authors use P53, while elsewhere in the paper the authors use p53. In line 91, should that be wild type p53, and JMURTK2 should be JMU-RK2, consistent with its designation elsewhere in the paper.  Ellipticine is misspelled in line 120.

5) Text lettering within the figures (e.g., scramble, RPL#1, RPL#2, scan measurement numbers, etc.) need to be larger.  They are too small right now.

Author Response

1) I have to admit I get lost in the descriptions of Figure 3.  The Fig. 3 legend begins by saying “Reduced nucleolar stress response suppresses topoisomerase inhibitor mediated activation of the p53 pathway…”  Should this read “Enhanced nucleolar stress response suppresses inhibitor mediated activation of p53…”? The nucleolar stress is induced by depleting RPL11.  When more RPL11 is depleted (enhancing nucleolar stress), there is a corresponding loss of p53 (a suppression of the p53 activation pathway).  If this is the correct interpretation, then the text and Fig. 3 legend need to be re-written to make it more clear.

We wish to express our gratitude to Reviewer#2 for the valuable suggestions. RPL11 depletion causes reduced levels of the mature 28S rRNA and accumulated 12S precursor rRNA, showing that RPL11 regulates ribosomal processing (Robledo S et al., RNA. 2008, 14(9)1918-29). Therefore, dysregulation of rRNA synthesis via RPL11 depletion may be expected to induce nucleolar stress response. However, several reports indicated that RPL11 depletion alone does not activate the p53 pathway and induce cell cycle arrest, suggesting that p53-dependent nucleolar stress response may not be induced by RPL11 depletion alone in a certain cell type (Pelletier J et al., EMBO J. 2020, 39(13)e103838; Bhat K.P et al., EMBO J. 2004, 23(12)2402-12; Sasaki M et al., Nat Med. 2011, 17(8)944-51; Lindström M.S et al., PLoS One. 2010, 5(3)e9578; Sloan K.E et al., Cell Rep. 2013, 5(1)237-47). Indeed, our western blotting analysis showed no significant activation of p53 pathway upon RPL11 depletion alone without drug treatment, suggesting that p53-dependent nucleolar stress response was not activated upon RPL11 depletion (Figures 3, 4, 5). Instead, RPL11 is shown to possess extra-ribosomal function as follows: Releases RPL11 from the nucleolus; binds and inhibits MDM2 upon nucleolar stress. Many studies using RPL11 siRNA demonstrated that RPL11 depletion weakens RPL11-mediated nucleolar stress response (Morgado-Palacin L et al., Carcinogenesis. 2014, 35(12)2822-30; Nishimura K et al., Cell Rep. 2015, 10(8)1310-23; Donati G et al., Cell Rep. 2013, 4(1)87-98; Sasaki M et al., Nat Med. 2011, 17(8), 944-51). Therefore, we concluded the results in Figure 3 as follows: “Reduced nucleolar stress response suppresses topoisomerase inhibitor-mediated activation of the p53 pathway in malignant rhabdoid tumor cells.”

2) RPL11 is known to interact with Myc (e.g., Olausson et al., Cells 2012 1, 774-798; doi:10.3390/cells1040774), so could the effects of depleting RPL11 be related to enhanced Myc activities?  The western blots in Fig. 3, 4, and 5 should also show Myc and perhaps protein products expressed from known Myc target genes.

We thank the reviewer for their insightful suggestion. As the reviewer mentioned, RPL11 regulates p53-independent nucleolar stress response via Myc activity. Since it is an important theme, we have planned to investigate whether topoisomerase inhibitors may affect Myc activity via RPL11 binding. We have addressed this issue in the Discussion section on page 9, lines 267-268, citing the relevant literature.

3) With the depletion of RPL11, is there an adverse effect on overall ribosome function and protein synthesis in general?

We thank the reviewer for the important question. As the reviewer mentioned, RPL11 depletion was shown to cause decreased ribosome production via rRNA processing dysregulation, resulting in reduced ribosome content and translational capacity (Teng T et al., Mol Cell Biol. 2013, 33(23)4660-7). Although the defect of ribosome function upon RPL11 depletion is observed, RPL11 depletion alone does not significantly affect p53 and its downstream target p21 (Pelletier J et al., EMBO J. 2020, 39(13)e103838). Again, these observations suggested that RPL11 might not influence p53 regulation via nucleolar stress response. In addition, MTT assay revealed that RPL11 depletion did not significantly alter MDM2 inhibitor sensitivity (Figure 1D, 2D), showing that drug sensitivity unrelated to nucleolar stress response is not affected by RPL11 depletion alone. Although ribosome function may be affected by RPL11 depletion at least in part, considering these observations and other studies that included RPL11 knockdown, our experimental conditions using RPL11 depletion seems ideal for measuring drug sensitivity and p53 activity via nucleolar stress response.

4) There are several minor discrepancies. For example, lines 89 and 90 and again in 119 and 120 the authors use P53, while elsewhere in the paper the authors use p53. In line 91, should that be wild type p53, and JMURTK2 should be JMU-RK2, consistent with its designation elsewhere in the paper.  Ellipticine is misspelled in line 120.

We are grateful to the reviewer for bringing this to our attention. We have corrected “P53” as “p53” throughout the revised manuscript and figures. Since the cell line name JMU-RTK-2 is used in the report that the cell line is first established from malignant rhabdoid tumor patient, we have corrected this as well throughout the revised manuscript. We have corrected ellipticine in the Results on page 4, line 125.

5) Text lettering within the figures (e.g., scramble, RPL#1, RPL#2, scan measurement numbers, etc.) need to be larger.  They are too small right now.

We apologize for the small lettering within the figure. To alleviate the reviewer’s concern, we have changed the lettering in the revised figures.

Round 2

Reviewer 1 Report

Dear authors thank you for addressing my comments and suggestion. The edits has improved the manuscript.  

Two recommendation:

1) Instead of adding numbers under each WB band, please provide an extra quantification bar graph

2) To clarify the number of replicates:   please change "Results represent independent multiple experiments." to n=number of replicates    

Author Response

1) Instead of adding numbers under each WB band, please provide an extra quantification bar graph 

We wish to express our gratitude to Reviewer#1 for their valuable suggestions. We applicate with the reviewer’s suggestion. As requested by the reviewer, we have added data with quantification bar graphs in figure R4-6 for the reviewer.

 2) To clarify the number of replicates:   please change "Results represent independent multiple experiments." to n=number of replicates

Thank you for your comment. We apologize for the missing description of the number of replicates in each experiment. We confirmed reproducibility via more than three independent experiments. Number of replicates of each experiment is followed.

Figure 1; Five experiments

Figure 2; Five experiments

Figure 3; Four experiments

Figure4; Four experiments

Figure 5 (A), (B); Three experiments

Figure 5 (C); Three experiments

Now we have added the numbers of replicates of each experiment. Please refer to page 3, line 117; page 4, line 124; page 5 line 150; page 5 lines 160; page 6, line 184.

Reviewer 2 Report

I have a better understanding of what the authors mean by “reduced nucleolar stress suppresses topo inhibitor-mediated activation of the p53 pathway”.  Simply, the authors use the depletion of RpL11 to suppress the p53-mediated response of nucleolar stress initiated by topoisomerase inhibitors.

But with ribosome life-span being 3 days, my concern remains that depleting RpL11 over a three-day period (Figures 1, 2 and 5A and B) could cause ribosome failure that then reduces p21 production for cell cycle arrest or Bax production for the onset of apoptosis.  We have to assume that the 0 uM drug cell viability controls for Figures 1, 2, 5A, and 5B infer no adverse effects on ribosome assembly or protein synthesis due to the loss of RpL11 over this three-day period. The immunoblots in Figures 3, 4, and 5C are not adequate controls for these cell viability assays because these immunoblots were prepared after only 24 hours of cell culture.  Would the immunoblots show the same results after 3 days of culture?

The paper is difficult to follow.  For one of many examples, Lines 97-99: “The sensitivity of JMU-RTK-2 cells to siremadlin was not significantly affected by RPL11 knockdown.” Meaning what?  Help the reader by briefly stating the biological significance of the observation.

I was disappointed to see that no experiments in the revision to address possible effects of RpL11 loss on Myc function.  Perhaps this is a future study.

Author Response

1)But with ribosome life-span being 3 days, my concern remains that depleting RpL11 over a three-day period (Figures 1, 2 and 5A and B) could cause ribosome failure that then reduces p21 production for cell cycle arrest or Bax production for the onset of apoptosis. We have to assume that the 0 uM drug cell viability controls for Figures 1, 2, 5A, and 5B infer no adverse effects on ribosome assembly or protein synthesis due to the loss of RpL11 over this three-day period. The immunoblots in Figures 3, 4, and 5C are not adequate controls for these cell viability assays because these immunoblots were prepared after only 24 hours of cell culture.  Would the immunoblots show the same results after 3 days of culture?

We wish to express our gratitude to Reviewer#2 for the valuable suggestions again. p53 signaling is known to be activated very quickly (~24hr) and then its effects are largely attenuated by feedback mechanism. Considering the effect on p53 signaling via the drug in the chronic phase, the effect of such feedback effects had to be taken into account, which could lead to complex interpretations. Thus we decided to examine the effect of p53 signaling via RPL11-mediated nucleolar stress response induced by the topoisomerase inhibitors until 24hr in the present study.

2)The paper is difficult to follow.  For one of many examples, Lines 97-99: “The sensitivity of JMU-RTK-2 cells to siremadlin was not significantly affected by RPL11 knockdown.” Meaning what?  Help the reader by briefly stating the biological significance of the observation.

We apologize for the inadequate description of the text. Since the siremadlin is an MDM2 inhibitor and its direct inhibition of MDM2 is known to activate p53 signaling independent of nucleolar stress. Thus the siremadlin is a negative control drug that activates p53 pathway and reduces the cell survivability regardless of RPL11-mediated nucleolar stress response. it is expected that reduced nucleolar stress response via RPL11 knockdown would not alter p53 pathway activation and drug sensitivity by this drug. We have addressed this issue in the Result section on page 2-3, lines 96-100.

3)I was disappointed to see that no experiments in the revision to address possible effects of RpL11 loss on Myc function.  Perhaps this is a future study.

We thank the reviewer for the insightful suggestion. We agree that myc signaling is very important to understand the mechanism of nucleolar stress response without p53. We have not yet established experimental conditions and evaluation methods for detecting myc signals, unfortunately, we do not have time to examine this issue at this time. Since myc signaling could conceivably be involved in the RPL11-mediated nucleolar stress response via the topoisomerase inhibitors, we aim to evaluate the effect of myc singinaling in a future study.